# Data-Driven Phenotyping of Alzheimer’s Disease under Epigenetic Conditions Using Partial Volume Correction of PET Studies and Manifold Learning

**DOI:** 10.3390/biomedicines11020273

**Published:** 2023-01-19

**Authors:** Silvia Campanioni, José A. González-Nóvoa, Laura Busto, Roberto Carlos Agís-Balboa, César Veiga

**Affiliations:** 1Cardiovascular Research Group, Galicia Sur Health Research Institute (IIS Galicia Sur), 36213 Vigo, Spain; 2NeuroEpigenetics Laboratory, Instituto de Investigación Sanitaria de Santiago (IDIS), Área Sanitaria de Santiago de Compostela-Hospital Clínico Universitario de Santiago (CHUS), 15706 Santiago de Compostela, Spain; 3Movement Disorders Group, Health Research Institute of Santiago de Compostela (IDIS), Área Sanitaria de Santiago de Compostela-Hospital Clínico Universitario de Santiago (CHUS), Servizo Galego de Saude-Universidad de Santiago de Compostela (SERGAS-USC), 15706 Santiago de Compostela, Spain; 4Neurology Service, Área Sanitaria de Santiago de Compostela-Hospital Clínico Universitario de Santiago (CHUS), 15706 Santiago de Compostela, Spain

**Keywords:** PET, Partial Volume Correction (PVC), manifold learning, artificial intelligence, data-driven AD phenotyping

## Abstract

Alzheimer’s disease (AD) is the most common form of dementia. An increasing number of studies have confirmed epigenetic changes in AD. Consequently, a robust phenotyping mechanism must take into consideration the environmental effects on the patient in the generation of phenotypes. Positron Emission Tomography (PET) is employed for the quantification of pathological amyloid deposition in brain tissues. The objective is to develop a new methodology for the hyperparametric analysis of changes in cognitive scores and PET features to test for there being multiple AD phenotypes. We used a computational method to identify phenotypes in a retrospective cohort study (532 subjects), using PET and Magnetic Resonance Imaging (MRI) images and neuropsychological assessments, to develop a novel computational phenotyping method that uses Partial Volume Correction (PVC) and subsets of neuropsychological assessments in a non-biased fashion. Our pipeline is based on a Regional Spread Function (RSF) method for PVC and a t-distributed Stochastic Neighbor Embedding (t-SNE) manifold. The results presented demonstrate that (1) the approach to data-driven phenotyping is valid, (2) the different techniques involved in the pipelines produce different results, and (3) they permit us to identify the best phenotyping pipeline. The method identifies three phenotypes and permits us to analyze them under epigenetic conditions.

## 1. Introduction

Alzheimer’s disease (AD) is a progressive neurodegenerative disorder associated with cognitive decline and is the most common form of dementia [1]. In a social context of progressive aging in Western countries, and estimations of the population affected by AD being above tens of millions worldwide, expected to quadruple over the next coming decades, it becomes a serious and major public health concern due to its increasing prevalence, chronicity, caregiver burden, and the high personal and financial costs of care [2]. Moreover, there are currently no treatments for AD that have been proven effective [3], and consequently, there is an urgent need to develop new strategies for the management of this new pandemic [1]. On the other hand, in recent years, several trials have demonstrated the efficacy of new treatments for AD in a well-selected subset of patients [4]. In such a scenario, precise phenotyping for identifying such patients plays a crucial role. This points directly to the AD phenotyping problem that could provide robust mechanisms for the identification of potential patients who could take advantage of those new treatments. Recent advances in methodologies for data characterization have introduced the data-driven phenotyping of AD and related dementias [4]. Such a computational approach is automated and non-biased, high-throughput, and can handle vast amounts of noisy healthcare data [5,6,7]. Phenotyping has been widely used in genetic studies, but in recent years, the concept has been extended to wider sources of data [8].

During the last decades, an increasing number of studies have confirmed epigenetic changes in AD, and although it is not clear whether these epigenetic changes are the cause or result of AD, they pave the way for new medical research worldwide [9]. Consequently, a robust phenotyping mechanism must take into consideration the environmental effects on the patient during the process. At the same time, extensive evidence has been produced that indicates that the pathological signs of AD may be amyloid plaques and neurofibrillary tangles [10], even if the underlying disease mechanism remains unknown. Imaging the brain for AD characterization has been applied extensively. The most employed technique for imaging the amyloid plaques is based on a combination of Positron Emission Tomography (PET) and Magnetic Resonance Imaging (MRI) [10]. These point directly to the purpose of this work: to set up a methodological pipeline to obtain AD phenotypes using hyperparameter optimization for the different components involved in the pipelines for the PET imaging of amyloid-β (Aβ) plaques using Partial Volume Correction (PVC) and manifold learning.

Tracers such as [11C] PiB, [18F-AV45] Florbetapir, and [18F] AV-4517 are used to quantify in vivo fibrillar Aβ decomposition, one of the main indicators of AD pathology [10]. The low spatial resolution of PET images reduces the indicator efficiency [11]. This resolution ranges from 5 to 6 mm [12] and it is quantified as Full Width at Half Maximum (FWHM). Due to this low resolution, the intensity of a particular voxel also reflects the tracer concentration of the surrounding area, not only the analyzed tissue area, complicating, even more, the quantification of Aβ plaque decomposition. In order to overcome the mentioned inconveniences of PET, due to low resolution and noise, the PET study must be merged with an MRI as a source of information that provides the anatomical information that is missed. Raw PET image-based quantification provides an inaccurate representation of Aβ deposition without adequate Partial Volume Correction (PVC), but there is no consensus regarding the correction processing pipeline to use [13]. In addition to PET studies, MRI has also been extensively employed to assess AD. Even if MRI usage is limited by factors such as imaging hardware, scanning time, and costs, this imaging modality plays an important role in clinical and brain exploration [14]. Previous works have discussed its role in cognitive impairment assessment from different approaches, such as using functional MRI, acquiring high-resolution MRI images clinically using super-resolution [15], or how brain MRI to PET synthesis can improve AD diagnosis. In this paper, T1-weighted Magnetic Resonance Imaging (MRI—T1W) studies are employed to build a brain atlas for each patient. 

At the same time, during the last decades, the scientific community has witnessed the emergence of artificial intelligence as an invaluable technique and tool to support research [16]. Within this broad field, manifold learning has emerged as a disciple that permits one to reduce the magnitude of data. Manifold learning can be thought of as an attempt to generalize linear frameworks such as Principal Component Analysis (PCA) and Linear Discriminant Analysis (LDA) to be sensitive to non-linear structures in data [17]. Many machine learning methods rely on the implicit assumption that data has some inherent structure. Manifold learning assumes that the observed data are found in a low-dimensional manifold embedded in a higher-dimensional space [17]. This assumption states that the shape of the concerned data is relatively simple.

During the last years, a lot of epidemiological research [18] has been suggesting that the simultaneous interactions of different lifestyle habits and environmental factors with Apolipoprotein E (APOE) alleles affect the risk of AD development. In the list of suggested epigenetic conditions repeated in the studies, the practice of physical exercise, nutritional habits, with a special focus on fat intake and ketogenic diets, coffee consumption, alcohol intake, education level, traumatic brain injury, cigarette smoking, and exposure to sunlight and pesticides have been gaining increasing attention. Even if the current level of evidence shows that younger APOE4 carriers in early stages would derive more benefit from preventive lifestyle interventions than older APOE4 noncarriers in later stages of dementia who may show the most pronounced effects, it is not robust. The current weak evidence or inconsistency could be caused by some confounding factors, such as sample sizes, the methodology used, and the demographic characteristics of the participants, including age and gender, potential related exposures, and comorbidities.

In this paper, a new methodology to identify AD phenotypes by selecting the best phenotyping pipeline (a combination of PVC techniques for feature selection on PET images and manifold learning for dimensionality reduction) under several epidemiological conditions, is presented. Section 2 presents the required background for the work, namely PVC methods, the dataset, the manifold learning approaches, the k-means clustering algorithm, and the set of software packages. Section 3 presents the proposed methodology. The results of applying the methodology to a cohort of patients derived from the OASIS-3 dataset are provided in Section 4. The discussion and conclusion are provided in the final section.

## 2. Background

This section covers the fundamental background descriptions required for the implementation of the proposed methodology, namely the dataset employed in the analysis, the fundamentals of PVC methods, the fundamentals of the manifold techniques and clustering techniques employed in this work, and the set of software packages that permits us to run the experiments presented in the results.

### 2.1. Dataset (OASIS-3)

In this work, the Open Access Series of Imaging Studies 3 (OASIS-3) [19] was used. It includes MRI, PET, and clinical data from 1378 subjects ranging from 42 to 95 years old. It was developed by Washington University Knight Alzheimer Disease Research Center over 30 years. There are several kinds of subjects: cognitively normal adults (755) and subjects with cognitive issues (662). MRI and PET images are used to analyze the chronological evolution over several tissue types and to improve aging models and cognitive impairment. There are 2157 PET images, available with the post-processed files generated using the PET Unified Pipeline (PUP), and 2842 MRIs available in OASIS-3. Two radiopharmaceuticals were used to investigate Aβ deposits in the brain [19]: Pittsburgh Compound B ([11C] PIB or PIB) and Florbetapir [18F] (18F-AV-45 or AV45).

Clinical and cognitive assessments were completed in accordance with the National Alzheimer Coordinating Center Uniform Dataset (UDS) allowing for their combination with multiple AD open data projects. UDS assessments include family history of AD, medical history, physical examination, and neurological evaluation [20,21].

### 2.2. Partial Volume Correction (PVC) Methods

PET imaging suffers from deteriorating factors, such as low spatial resolution, poor signal-to-noise, and count-dependent bias. This imaging technique is relatively blurry compared to other imaging techniques, such as CT and MRI. Due to these factors, when the radioactivity within a region is measured, it results in distortion in the quantification of Aβ plaques. This distortion in the quantification of small-volume concentrations occurs due to a phenomenon known as the Partial Volume Effect (PVE). PVE causes the intensity of a particular voxel to reflect the concentration of the tracer of the tissue within that voxel and also of the surrounding area. The correction of this phenomenon is important because it can be a noise factor in between-subject or longitudinal clinical studies. Different correction techniques have been developed to mitigate the PVE, but none of them were optimal in all imaging scenarios [10].

Since the first attempts, back in the seventies of last century, to solve the inherent drawbacks of PET, a wide range of PVC techniques have been proposed for the correction of PVE in PET, with a special peak in the nineties of the last century. Figure 1 shows a historical schema of the development of these techniques over the years. In this work, two of these algorithms are used: the Müller-Gärtner (MG) method [22] and the Geometric Transfer Matrix (GTM) method, also called the Regional Spread Function (RSF) [23]. Meltzer’s method uses Partial Volume Correction with Two-Components (PVC2C), which defines two categories of tissue (brain and non-brain); but it corrects the PVE in non-brain tissue only. The MG PVC uses three components: gray matter, white matter, and cerebrospinal fluid. This method compensates for the effects caused by PVE in gray matter. The RSF method corrects the PVE between all brain regions and recovers the true activity for multiple regions [23]. It is based on a PVC algorithm that uses volumetric anatomical data from an MRI. The whole-body concentration of the injected radioactivity and the estimated regional Standard Uptake Value Ratio (SUVR) allow us to investigate the impact of PVC on the PET image quality of Aβ plaques [24]. There is still an open discussion of which is the best approach or not applying PVC corrections at all [10,13].

### 2.3. Dimensionality Reduction and Manifold Learning

Dimensionality reduction methods consist of mapping a dataset to a lower-dimensional subspace, obtained from the original space. Such methods allow us to express the data using fewer parameters, at a lower cost, which makes them very useful in machine learning [25]. Reducing high-dimensional datasets to two or three dimensions makes them much easier to analyze and visualize, as the data can be plotted and show the structure of the data in a more intuitive way. Another way to understand manifold learning is as a generalization of linear frameworks, such as PCA or LDA, but sensitive to non-linear data.

Although there are supervised manifold learning techniques, the most common problem is unsupervised, as it exploits the information in the data without any given labels.

There are several computational approaches and algorithms available for manifold learning. Some of the most popular implementations employed by the research community are described in the next subsections.

#### 2.3.1. t-Stochastic Neighbor Embedding (t-SNE)

t-SNE is a dimension reduction technique used for the exploitation of large dimensional data that has been developed in 2008 by Hinton and Maaten [26]. As in the case of linear methods, the purpose is to determine one space with a lower dimension while keeping the same distance between the points. The technique is a variation of Stochastic Neighbor Embedding [27], based on creating a probability distribution that represents the similarities between neighbors in a high-dimensional space and in a lower-dimensional space. By similarity, it converts the distances into probabilities. It is divided into three main steps:

First step: calculate the similarities of points in the initial high-dimensional space. For each point $x_{i}$, a Gaussian distribution around this point is produced. Then measure, for each point $x_{j}$ (i different from j), the density under this previously defined Gaussian distribution. Finally, normalize each of the points. Thus, a list of observed conditional probabilities is obtained as:(1)$$p_{ij}=\frac{\exp\left(\frac{-{\left|\left|x_{i}-x_{j}\right|\right|}^{2}}{2\sigma^{2}}\right)}{\sum_{k\neq l}\exp\left(\frac{-{\left|\left|x_{k}-x_{l}\right|\right|}^{2}}{2\sigma^{2}}\right)\;\;\;\;}$$

Standard deviation s is defined by a value called *perplexity* that corresponds to the number of neighbors around each point. This value is set previously and allows for estimating the standard deviation of the Gaussian distributions defined for each point $x_{i}$. The greater the perplexity, the greater the variation.

Second step: distribute the points randomly in the low-dimensional space. The rest is similar to the first step, calculating the similarities of the points in the newly created space, but using a t-Student distribution and not a Gaussian one. In the same way, it is possible to obtain a list of probabilities as:(2)$$q_{ij}=\frac{{\left(1+{\left|\left|y_{i}-y_{j}\right|\right|}^{2}\right)}^{-1}}{\sum_{k\neq l}{\left(1+{\left|\left|y_{k}-y_{l}\right|\right|}^{2}\right)}^{-1}}$$

Third step: in order to faithfully represent the points in the low-dimensional space, the similarity measures in both spaces should ideally match. So, we need to compare the similarities of the points in both spaces using the Kullback–Leibler (KL) distance. KL distance will be minimized by gradient descent to obtain the optimal point $y_{i}$ in a space of lower dimensions. This is equivalent to minimizing the difference between the probability distributions between the original space and the low-dimensional space.

#### 2.3.2. Isometric Mapping Embedding (ISOMAP)

Multidimensional Scaling (MDS) is the classical technique used to exploit the local linearity of manifolds and create a mapping that preserves local neighborhoods at each point of the underlying manifold. The input data used in manifold learning are sampled from a low-dimensional manifold that is embedded within a higher-dimensional vector space.

ISOMAP [28] is based on the Floyd–Warshall algorithm and MDS. It takes a matrix of pair-wise distances between points, assuming only neighboring points know the distances. Then, it computes the pair-wise distances between the remaining points by using the Floyd–Warshall algorithm, estimating the matrix of pair-wise geodesic distances between all the points. Afterward, ISOMAP reduces the dimension of the points’ positions using classic MDS.

#### 2.3.3. Uniform Manifold Approximation and Project Embedding (UMAP)

UMAP is a novel algorithm for dimension reduction that can be used for visualization and general non-linear dimension reduction [29]. UMAP first builds a high-dimensional representation of the data after deriving a lower-dimensional graph and optimizing the similarity in their structures. To determine connectedness, the algorithm extends a radius from each point and, when those radii overlap, UMAP connects the points. Setting the value of the radius is a critical decision, as too-low values would lead to small, isolated clusters, whereas high values would connect everything. To overcome this issue, UMAP chooses the radius locally, based on the distance between each point and its nth nearest neighbors [24].

### 2.4. Clustering Technique, k-Means

k-means is an unsupervised classification (clustering) algorithm that groups objects into k groups based on their common mean values [30]. The grouping is performed by minimizing the sum of distances between each object and the centroid of its group or cluster, using very often the quadratic distance for this purpose. Computationally, the algorithm sets the position of the centroid of each group, taking as the new centroid the position of the average of the objects belonging to said group. This means solving an optimization problem, being the function to optimize the sum of the squared distances of each object to the centroid of its cluster. Objects are represented as vectors on a D-dimensional hyperspace (x1,x2,…,xn) and the k-means algorithm identifies *k* groups where the sum of distances of the objects is minimized, within each group $S=\left\{S1,S2,\;\ldots ,Sk\right\}$, to its centroid. The problem can be formulated as follows:(3)E S  minμi=  S  min ∑i=1k∑ xjϵSixj−μi2
where *S* is the dataset, whose elements are the objects $x_{j}$ represented by vectors of features and the operator ${\left|\left|x_{j}-\mu_{i}\right|\right|}^{2}$ the Euclidean distance, minimizing the sum of the square of the distances between each data point and its centroid within a cluster. The algorithm provides k-clusters each with a centroid $\mu_{i}$.

k-means also provides a way to measure the internal cohesion within clusters and the external separation between clusters. Those measurements are based on an intra-cluster variance and an inter-cluster variance. The inter-cluster *WCSS* (Within Clusters Sum of Squares) accounts for the total variations within a cluster.
(4)$$WCSS=\overset{k}{\underset{i=1}{\sum }}\underset{\;x_{j}\epsilon S_{i}}{\sum }{\left|\left|x_{j}-\mu_{i}\right|\right|}^{2}$$

$\mu_{i}$ is the centroid of each cluster; in other words, the minimum value found in the minimization process of Equation (3).

An inter-cluster, the sum of squares between (*SSB*), could be used to quantify external separation. It is defined as the sum of the squared distance between the global average point and each centroid. The bigger the value, the better the clustering:(5)$$SSB=\sum_{i=1}^{k}{\left|\left|\mu_{i}-ga\right|\right|}^{2}$$
where ga is the global average coordinates of the whole dataset. Both Equations (4) and (5) allow us to analyze the variability in terms of *k* values, *elbow* curves, and silhouette plots, which display a measure of how close each point in one cluster is to points in the neighboring clusters, assessing visually the number of clusters.

### 2.5. Software Packages and Tools

To process the images, as described in Section 2.2, and obtain the derived features employed on the method described in this paper, a set of software packages are required; namely, PET images were preprocessed using the PET Unified Pipeline [31]. Single MRI—T1W images were processed through the software Freesurfer v7.2.0 [32], a package that permits one to provide volumetric MRI data and segmentations maps. These maps are used to determine cortical volumes or Regions of Interest (ROIs) for PET imaging. The Python 3.9 library employed for producing the graphs was Matplotlib [33] and for producing the manifold and k-means codes was Scikit-learn [34].

## 3. Method

This paper presents a new methodology for the identification of AD phenotypes by analyzing the PET images of AD patients under epigenetic conditions. The method is based on the integration of two main parts, in a single phenotyping pipeline (PPi), and its optimization. Pipelines include a combination of several PVC approaches for extracting SUVR features from PET images and several manifold learning approaches for dimensionality reduction. The overall schema of the method is provided in Figure 2, which presents those main blocks.

The first part deals with the obtention of different PET-corrected studies. In our case, we propose to use three different methods, namely RSF, PVC2C, and noPVC (not applying PVC), all described in Section 2.2, and the extraction of regional features from the corrected images using the MRI-derived Desikan–Killiany atlas. This part is carefully described in Section 3.1. After PVC, PET images are processed to extract the set of features (SUVR values for each Region of Interest (ROI)), producing information that will be provided to the next block of manifold learning in the second part of the pipeline. That part will produce a nonlinear reduction in dimensionality, using three different state-of-the-art methods, namely t-SNE, UMAP, and ISOMAP (all described above, in Section 2.3). This process will be better described in Section 3.2. Finally, all the results generated by the different phenotyping pipelines will be gathered on a common evaluation table by using the k-means algorithm, which permits us to merge the outputs in a unified way, and consequently compare the results from the mix of different combined approaches, providing the best set of clusters, identified as phenotypes.

To simplify the method explanation, PPi-s are named PP1 (combining PVC2C and UMAP), PP2 (combining RSF and UMAP), PP3 (combining noPVC and UMAP), PP4 (combining PVC2C and t-SNE), PP5 (combining RSF and t-SNE), PP6 (combining noPVC and t-SNE), PP7 (combining PVC2C and ISOMAP); PP8 (combining RSF and ISOMAP), and PP9 (combining noPVC and ISOMAP).

### 3.1. Obtention of Imaging Derived Features after PVC

This part of the method deals with the obtention of the different SUVR values, obtained for all ROIs in the images by applying several PVC methods. The imaging pipeline has a double path, one for processing the PET studies and another for the MRI studies. The MRI path is devoted to obtaining the ROI to compute the SUVR values on the PET after PVC correction. For this purpose, the Desikan–Killiany atlas [35] was employed, Figure 3 shows the different stages of the MRI process [32]. The atlas color labels [35] are shown in Figure 3c,d.

In the other imaging path, PET images are corrected to achieve a common spatial resolution of 8 mm to minimize inter-scanner differences [36]. Even if the methodology is general and several methods could be considered, this work employs three of them. The first method is the PVC2C approach [37,38], which is the most widely represented in the amyloid imaging literature. The second method is based on the calculation of the RSF [23,39], which has also been widely applied in PET image analysis. The third approach is based on not applying PVC to the PET study, as in the literature there is no consensus regarding whether PVC is necessary or not for quantitative PET analysis [10,13]. Figure 4 shows the differences that PVC methods introduce in the PET image studies.

The vector–gradient algorithm [40] is used for symmetrical PET-MR registration. This method consists of an average transformation for PET > MR, and the inverse of MR > PET was used as the final transformation matrix. The Freesurfer software provides regional PET segmentation using wmparc.mgz for the definition of the regions. Figure 5 shows the result of the analyzed region. This method generates two reports: regional measurements and SUVR images.

The output of this stage is an array of 166 SUVR values. This array of values can be obtained for each patient study and for each correction method. This huge amount of data requires an approach to reduce the dimensionality and make it affordable to manage.

### 3.2. Obtention of Manifold Derived Clusters

The second stage of the pipeline is devoted to reducing the dataset dimensions and merging it into the optimal set of clusters. It has two main parts. The first part of the process deals with dimension reduction using the manifold methods. Those methods, as explained above in Section 2.3, use an implicit parameter (*D*), which is the final dimension of the dataset in the manifold. *D* has usually a very low value, and is much lower than the original dimension; in this work, the value of *D* will be determined heuristically, selecting the lowest value that permits us to construct the pipelines while maximizing the silhouette coefficient. The whole dataset of images is processed using each pipeline. After reducing the dimensionality of data, in the second part of this stage, data will be clustered to identify the phenotypes using k-means. k-means uses an implicit variable of the number of clusters to be identified (*k* in Equation (3)). After obtaining the clusters for each PPi, the metrics above-mentioned of the intra- and extra-cluster separability, as well as the p-values between clusters, are obtained. Data in this new format will permit us to identify the best number of clusters, using the elbow method and silhouette analysis. The elbow method [41] is the most popular for finding the optimal number of clusters; this method uses *WCSS* and *SSB* scores, which account for the total variations within a cluster using Equations (4) and (5).

## 4. Results

In order to evaluate the results that produce each PPi on the Alzheimer’s level in PET studies under epigenetic conditions, several experiments have been conducted. To assess statistically the evaluation, the OASIS-3 dataset (described in Section 2.1) was employed. A cohort of patients from OASIS-3 has been selected to fulfill the requirements of this work. A sub-dataset was extracted by selecting the cases with complete structural MRI—T1W, Aβ-PET scans, and clinical variables, which accounts for a total of 532 patients. The radiotracer injected for the PET image was the 18F-AV-45 for Aβ tracing. Statistics of the final cohort are provided in Table 1. The difference between the total number of cases and the sum of determinations of the variables is due to the unknown determinations of some patients. Statistical values are obtained only for the known values and no imputation methods are employed.

### 4.1. Hyperparameter Selection

The first set of experiments was conducted to identify the values of the hyperparameter that affects all pipelines (PPi), namely the final dimension of manifold (*D*) and the number of clusters in k-means (*k*). Concerning *D*, several analyses have been conducted to heuristically identify the optimal value. Using the PP5 and *k* = 3, the processing pipeline was applied for several *D* values, computing the averaged silhouette coefficients. As for *D* = 3, the average silhouette score is 0.3217. The result is worse than the average silhouette score of 0.563 obtained for *D* = 2, and the purpose of this stage is to reduce as much as possible while keeping the information to solve the problem (high values of *D* make data unmanageable). In the remainder of this work, pipelines were implemented considering *D* = 2. Concerning the determination of *k*, the nine PPi have been clustered using several values of *k* (in the range of 1–20). This permits us to produce the value of *WCSS* using Equation (4) and *SSB* using Equation (5) for each possible value of *k* (*elbow* analysis). Figure 6 presents the results of applying the silhouette analysis of the data for *D* = 2.

In this work, k-means was implemented using the classical Lloyd algorithm, using the following parameters: 10 as the number of times to run with different centroid seeds, 200 as the maximum number of iterations for a single run, and a value equal to 0.0001 for the relative tolerance with regards to the Frobenius norm of the difference in the cluster centers of two consecutive iterations to declare convergence. Figure 7 presents the curves for such analysis, where it can be seen that *k* = 3 is the optimal value of this parameter because the average silhouette score decreases as *k* increases. Such a value is the largest when *k* = 1.

To study the inter-cluster distances, we used silhouette analysis. Figure 6 presents the results of such an analysis for PP5, showing how close each cluster is to its neighbors and allowing a visual evaluation of parameters such as the number of clusters. In Figure 7, we can observe an elbow shape, where the graph has a sudden change to an asymptotic behavior on the X-axis. The k value in the elbow is the optimal *k* value, i.e., the optimal number of clusters, *k* = 3 in this case.

### 4.2. Aβ—PET Phenotypes Identification

The second set of experiments was conducted to evaluate the effects that each PPi produces on the emerging phenotypes. Using the image studies of those patients, as described above in Section 3.1, the features derived from the nine PPi (RSF, PVC2C, and noPVC) have been applied to the whole set of 532 PET studies from 487 patients. Figure 4 shows the output for the three approaches of PVC for one of the patients included in the study. In this work, we propose using 166 ROIs, as explained above. Different PVC methods provide different values of SUVR. Figure 8 presents the results of applying the methods employed in this work, RSF (in red), PVC2C (in blue), and noPVC (in purple), to the dataset above described for the 166 ROIs. In this figure, it is clear that the approaches have different results for all the ROIs. Consequently, choosing a correction method will have consequences for the derived phenotypes.

The results of applying the PVC methods provide a triple matrix file, with 166 rows and 532 columns, for the SUVR values on the 166 regions of each patient. This high-dimensional set will enter the manifold learning block using the pipeline described in Section 3.2, giving a result of the reduction in a two-dimensional (*k* = 2) set of variables that will be grouped into three clusters using k-means (*k* = 3), as explained in the previous section. After the dataset is processed using all pipelines (PPi), the next step is to identify the clusters in the data and consequently the phenotypes, named Phenotype1 (phen1), Phenotype2 (phen2) and Phenotype3 (phen3), using the methodology described in Section 3.2. In this way, metrics based on mathematical measurements can be easily obtained. The metrics employed for measuring the clinical variables of the phenotypes in this work were p-values of the different clusters that emerged (phen1 vs. phen2, phen2 vs. phen3, and phen1 vs. phen3), as well as the inter- and intra-cluster variance, as explained in Section 2.4, using Equations (4) and (5). All these data are provided in Table 2 for the nine PPi.

The results of this process are presented graphically in Figure 9, which presents the emerging clusters (now understood as phenotypes) of several manifold methods applied to the three PVC methods. In those nine scatter plots (a to i), the abscissa is the first component of the manifold reduction, and the ordinate is the second component of the manifold decomposition. In those images, the axis is different for each pipeline, because the results take values on different ranges for each method. According to the clustering results and the optimized cluster number, three clusters were identified.

Once the three clusters are identified for all PPi, the values of cognitive impairment were calculated by computing the mean value and the standard deviation of patients’ Mini-Mental State Examination (MMSE) included in each cluster, allowing the phenotype identification and characterization. The MMSE measures general cognitive status, ranging from 0 (severe impairment) to 30 (no impairment), based on a 30-point questionnaire that is used extensively in clinical and research settings [19]. The results for each PPi are provided in Table 2. Furthermore, this table also provides the p-value of the inter-phenotype comparisons, using the one-way analysis of variance (ANOVA) test, as well as the intra-cluster and inter-cluster measurements, using Equations (4) and (5).

In this way, the method permits us to identify the clearest phenotypes as the best-separated clusters. From those results, we can deduce that the best results are from line5, generated by PP5 (based on the t-SNE and RSF methods), providing values of *WCSS* = 26.029 and *SSB* = 19.341 and intra-cluster p-values of 0.012, 0.093, and 0.000, respectively, providing the total lowest values on average. More detailed results of pipeline five are provided in Table 3. Table 3 provides the statistical values using the clinical variables for the three phenotypes that have emerged in the previous step that identify PP5.

Analyzing the results in the table, considering gender, phen1 is well balanced, while in phen2 and phen3 the number of females is almost double that of men. Concerning age, the phenotypes are clearly separate: phen3 is the youngest at about 62.1 years old, phen1 is in the middle with a mean of 67.3 years, and phen2 is clearly different and the oldest group at 73.5 years; all values are means. Concerning cognitive impairment, the MMSE shows lower values correlated with age, moving from a nominal value of 29.3 in phen3, ranging to 28.280 in phen2, the oldest phenotype.

### 4.3. Epigenetic Analysis

In order to explore the way in which the proposed methodology can be employed to analyze how environmental conditions affect the emerging phenotypes, the above-identified phenotyping pipeline PP5 has been applied, repeating the analysis process with different environmental variables—such as the number of education years; TOBAC, measuring the binary smoking condition as cigarette smoking history with respect to the subject smoking more than 100 cigarettes in his/her life; DEPOTHR, measuring the occurrence of depression or other episodes during the previous 2 years; and alcohol abuse, as clinically significant impairment over a 12-month period manifesting in work, driving, legal, or social spheres—confronted with APOE. Results are provided in Table 4.

Education year is presented as the mean and standard deviation; TOBAC is presented as a binary condition. DEPOTHR: Depression or other episodes prior to 2 years. APOE genotype—low risk: E2/E2, E2/E3, E3/E3; medium risk: E2/E4, E3/E4; high risk: E4/E4. Substance abuse: alcohol.

The figures in the table reveal that the environmental effect is very different on the phenotypes. Particularly, we can see that three phenotypes are characterized differently than in the previous subsection, which indicates that including epigenetic conditions in the analysis provides different results, as should be supposed. The level of education, in terms of the number of education years, is similar, with mean values ranging from 15.898 to 16.247, phen3 being the group with the highest level of education and lowest variability within the group. The smoking condition is very different, where in phen2 there is a higher proportion of smokers (more smokers than nonsmokers) at 58/38, and in phen1 and phen3 the proportion is 73/88 and 73/107, respectively. The psychiatric condition is affected differently; remarkably, phen3 is the group that has a smaller number of cases, 5 in 183, doubling the ratio with the other groups. Concerning alcohol consumption, the first remark is on the low value of cases; in this cohort study only a few cases are present—14 subjects in total—and, consequently, conclusions are difficult to extract.

Upon obtaining the results, the “violin” plots can be employed to visually explore the different phenotypes. The results are presented in Figure 10. It is important to remark that, in this study, except for numerical values, all results are categorical and do not permit us to extract useful insights. The results of this analysis allow us to identify the best choice for the set of combinations proposed in the methodology.

From Figure 10, it is possible to conclude that phen2 presents higher values (mean) for age and years of smoking, while phen3 is the one with the lowest values for these parameters. Regarding the MMSE and the years of study, phen3 has higher values, while phen2 presents the lowest values. Phen1 presents intermediate values compared with phen2 and phen3. Therefore, phen3 could be associated with cognitively healthy people, phen1 with people with questionable dementia, and phen3 with mild or higher cognitive impairment.

## 5. Discussion and Conclusions

The objective of this study was to develop and validate a phenotyping method to test the hypothesis that there are identifiable AD phenotypes that are based on hidden patterns of the SUVR values of the PET studies of cognitive functions and epigenetic conditions. The presented phenotyping method uses pipelines of PVC and manifold methods. The proposed methodology uses both neuroradiological features from imaging and neuropsychological features from cognitive tests in an epigenetic environment. We compared the interpretability and discriminability of the phenotyping method. We compared the interpretability and discriminability of the phenotyping method using k-means.

Using this method, we derived three phenotypes. The method also identifies the hyperparameters (*D* and *k*) that play an important role in the results. This work presents the results of OASIS3 patients using nine PPis in a cross-validation matrix of three PVC methods and three manifold methods. The method could be easily extended to a larger number of PVC techniques and manifold methods without losing generality.

The results presented in this work show that PVC methods produce a different set of derived features (as shown in Figure 8), and consequently, the selection of the method will play an important role in any other derived usage of the data. Once the features are extracted from the dataset, there are several manifold approaches that allow us to reduce the dimensionality of the data and consequently manage the information in a more suitable way. The proposed methodology allows us to determine which combination of components of the processing pipeline is the best for each approach to the epidemiologic analysis using a mathematical formulation based on intra- and extra-cluster variabilities (Equations (4) and (5)). The method involves several parameters, beyond the manifold algorithm, that must be selected as the number of the final dimension (*D*) and the number of clusters to be identified in the cluster reassembling (*k*). A procedure that permits us to identify those parameters was presented (elbow and silhouette analysis). The experiments conducted in this work permit us to identify those values for the dataset. Particularly *D* = 2 and *k* = 3 are the optimal parameters for analyzing PET images under epigenetic conditions.

The methodology proposed in this paper was employed to compare the methods PVC2C, RSF, and noPVC, and the output of the processed data was employed to feed the manifold pipeline to compare the effects of the manifold method on the aggregability of patients under different clusters. This allows us to measure the separability of the clusters using p-values between the outputs for each output variable involved in the analysis (in this work, the following variables have been employed: age, years of study, drinking, smoking, psychiatric events, and APOE), including epigenetic and non-epigenetic conditions. The proposed methodology allows us to identify the best set of phenotyping pipeline components involved in the analysis. Using the patients described in this work, the best results appear (PP5) using RSF as a PVC method and t-SNE as a manifold method, with an inter-phenotype separability (*WCSS* = 26.029), an extra-phenotype separability (*SSB* = 19.341), and a set of p-values of 0.012, 0.093, and 0.000 for phenotypes one and two, two and three, and one and three, respectively. The three phenotypes are mainly characterized considering gender: phen1 is well-balanced, while in phen2 and phen3 the number of females is almost double that of men. Concerning age, the phenotypes are clearly separate: phen3 is the youngest at about 62.1 years old, phen1 is in the middle with a mean of 67.3 years, and phen3 is clearly different and the oldest group at 73.5 years; all values are means. Concerning cognitive impairment, the values of MMSE, which shows lower values correlated with age, move from a nominal value of 29.3 in phen3 to 28.280 in phen2, the oldest phenotype.

Using this method, the derived phenotypes also characterize the epigenetic variables using violin plots to visualize the outcomes, and, consequently, how patients could decline in cognitively distinguishable ways. Phen2 could be associated with cognitively healthy people, phen1 with people with questionable dementia, and phen3 with mild or higher cognitive impairment.

The inclusion of other databases, either public (such as the Alzheimer’s Disease Neuroimaging Initiative, ADNI) or private (from any research group that could provide it), could be a way to provide more robust results and strengthen the validity of the proposed method. It is important to take into consideration the need for having simultaneously both types of imaging studies (PET and MRI) and the clinical variables employed in the studio. This is an additional analysis that lies out of the scope of this paper and is a clear new future work line. The methodology could also be extended by including longitudinal data on the pipelines, to permit the characterization of the longitudinal progression of AD by understanding the neurodegeneration patterns. Another future line of research to explore is to evaluate how the proposed methodology could be extended and made more general by including additional evaluation metrics in the analysis to further demonstrate the effectiveness of the method (including measures of accuracy, precision, recall, F1 score, mean squared error, or the coefficient of determination). Presenting a diverse range of evaluation metrics could provide a more comprehensive assessment of the method’s performance. The main issue to solve is that the above-mentioned metrics involved in measuring artificial intelligence classifiers are not eligible as they require a gold standard value in order to build the confusion matrix elements (TP, TN, FP, and FN). In this methodology, there is no a priori true phenotype. This, again, lies out of the scope of this paper.

## Figures and Tables

**Figure 1 biomedicines-11-00273-f001:**
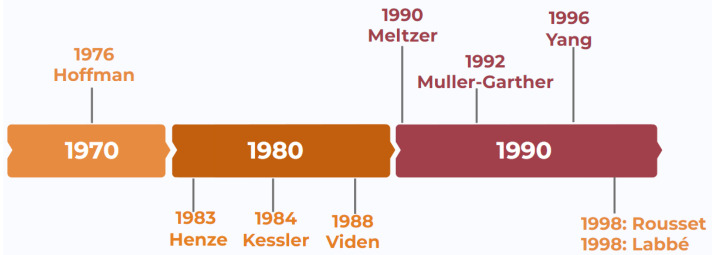
Historic chronogram of the presentation of several Partial Volume Correction (PVC) methods throughout the last decades of the last century.

**Figure 2 biomedicines-11-00273-f002:**
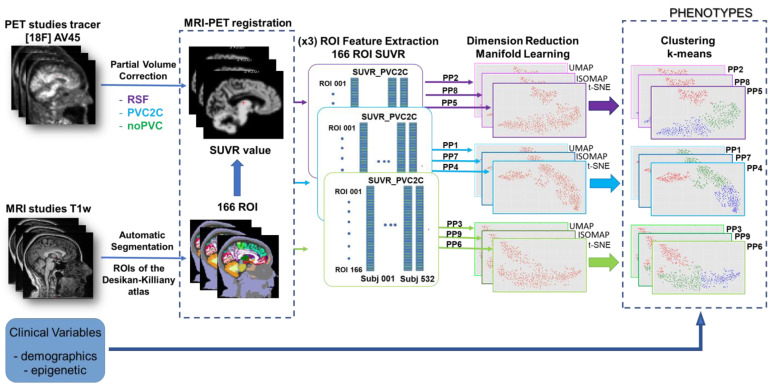
General schema for the method presented in this paper. Two different imaging modalities Positron Emission Tomography (PET) and Magnetic Resonance Imaging (MRI) are employed to extract the set of 166 Regions of Interest (ROIs). The ROIs are dimensionality-reduced by different methods (colored in green, blue, and purple lines in the schema), which will permit them to be clustered in the final stage to extract the phenotypes.

**Figure 3 biomedicines-11-00273-f003:**
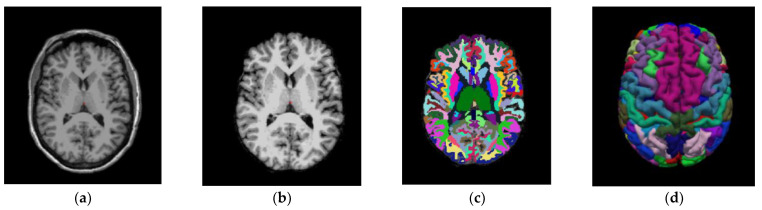
Coronal image of a patient from the study. (**a**) T1-weighted Magnetic Resonance Imaging (MRI—T1W). (**b**) MRI output after processing stages: Motion Correction and Conform, Intensity Normalization, Remove Neck and Skull Strip. (**c**) MRI output after applying Talairach transform computation and Cortical Parcellation Desikan-Killiany, an atlas that will permit us to obtain the values of Standard Uptake Value Ratio (SUVR) on the same patient from the PET image. (**d**) Three-dimensional reconstruction of the atlas from (**c**) using the color map of Desikan-Killiany atlas.

**Figure 4 biomedicines-11-00273-f004:**
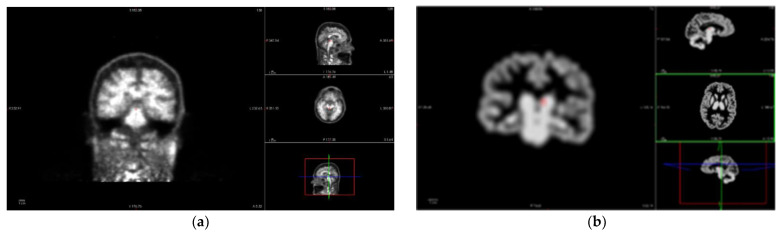
PET images of the same patient as Figure 3. (**a**) PET image before applying Partial Volume Correction (PVC). In axial, sagittal, and coronal planes and 3D reconstruction, clockwise. (**b**) Result of applying Partial Volume Correction with Two-Components (PVC2C) to the PET study in (**a**), displaying the same planes in the same order.

**Figure 5 biomedicines-11-00273-f005:**
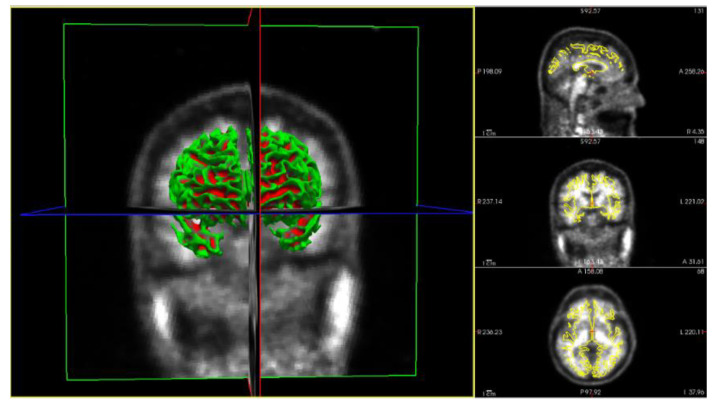
MRI-PET image registration after PVC of the same patient from Figure 3 and Figure 4, which will permit us to obtain the values of SUVR for the same patient, where different colors identify different MRI brain regions.

**Figure 6 biomedicines-11-00273-f006:**
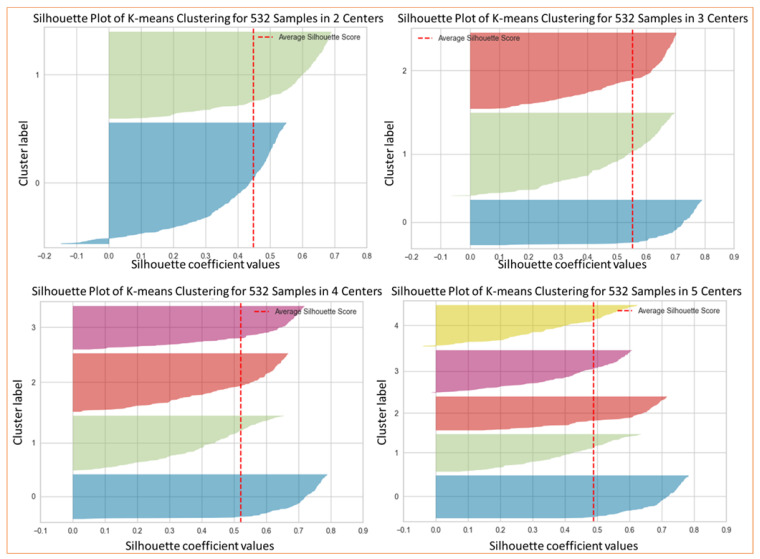
Values using the k-means of silhouette coefficients versus the number of clusters.

**Figure 7 biomedicines-11-00273-f007:**
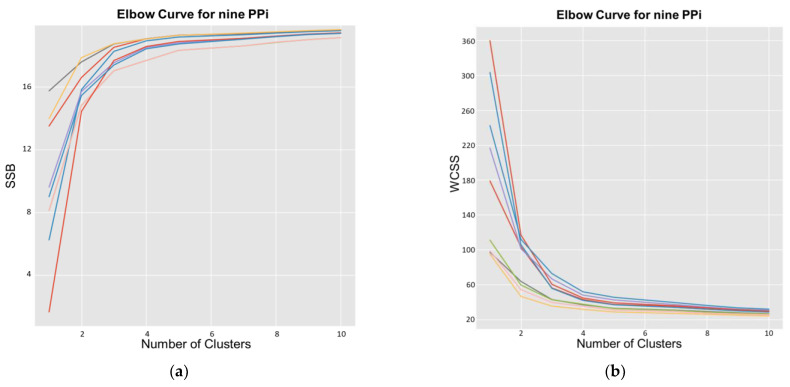
Values of score versus the number of clusters, using k-means for the elbow method. (**a**) Increasing elbow curves for the nine phenotyping pipeline (PPi) using *SSB* Equation (5). (**b**) Decreasing elbow curves for the nine PPi with respect to *WCSS* using Equation (4).

**Figure 8 biomedicines-11-00273-f008:**
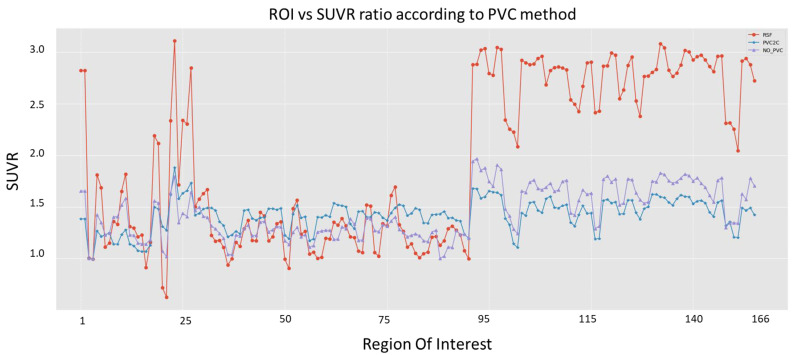
Plot of the mean SUVR values of the 166 anatomical regions obtained using the Desikan–Killiany atlas from the MRI study and after the application of the three PVCs to the original PET studio of 532 patients. In red, Regional Spread Function (RSF); in blue, PVC2C; and magenta, noPVC.

**Figure 9 biomedicines-11-00273-f009:**
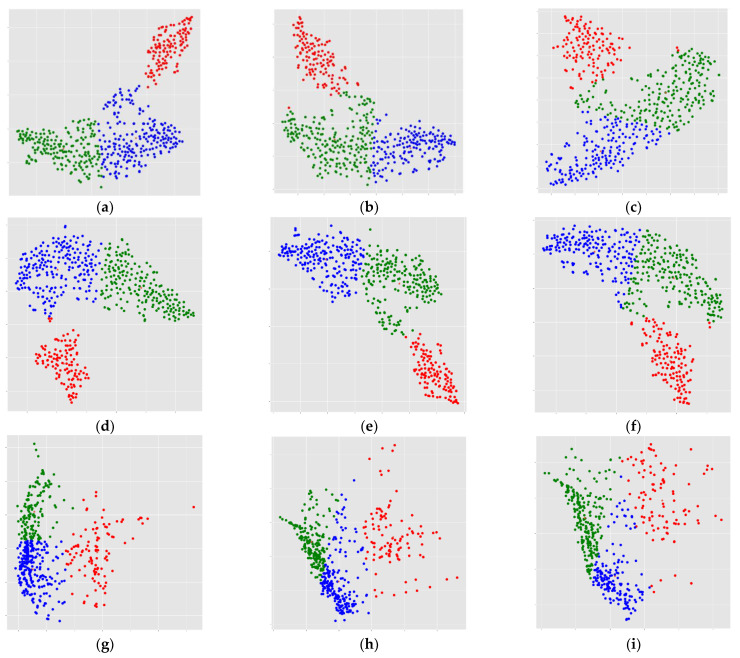
Graphical display of the 27 possible phenotypes that have emerged from the methodology proposed in this work, applied to the OASIS-3 dataset using 9 PPis and cross-correlated hyperparameters. The upper 3 graphs in a horizontal line (**a**–**c**) were obtained using UMAP; the three in the middle (**d**–**f**) were produced using t-SNE; and the three at the bottom (**g**–**i**) were produced using ISOMAP. The column on the left is for RSF, the column in the middle is for PVC2C, and the column on the right is for the noPVC correction. Cluster 1 is in red, cluster 2 is in green, and cluster 3 is in blue in all plots.

**Figure 10 biomedicines-11-00273-f010:**
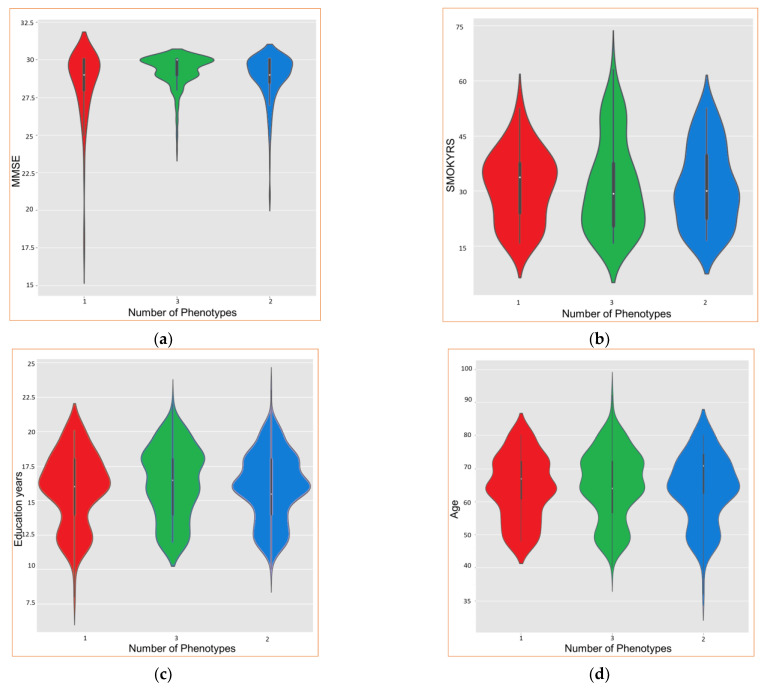
Epigenetics analysis with violin graphs. Differences between phen1, phen2, and phen3 in (**a**) MMSE, (**b**) smoking years, (**c**) education years, and (**d**) age. In red is phen1, in green is phen2, and in blue is phen3.

**Table 1 biomedicines-11-00273-t001:** Description of the OASIS-3 cohort of 532 sub-datasets included in this work.

Basic Demographics	Parameter	Statistical Description
Gender	Female	301
	Male	216
APOE	Low risk	316
	Moderate risk	166
	High risk	21
CDR	0	423
	0.5	67
	1	6
	2	2
TOBAC	Yes	194
	No	244
Hand	Left	37
	right	470
	Ambidextrous	10
Education years	Mean ± std	16.077 ± 2.511

APOE genotype: for E2/E2, E2/E3, E3/E3—low risk; for E2/E4, E3/E4—medium risk; for E4/E4—high risk. CDR: Dementia status was assessed for the UDS using the Clinical Dementia Rating17 (CDR) Scale, with CDR 0 indicating normal cognitive function; CDR 0.5: very mild impairment, questionable dementia; CDR 1: mild cognitive impairment; CDR 2: moderate dementia; CDR 3: severe cognitive impairment [21]. TOBAC: UDS question, cigarette-smoking history—has the subject smoked more than 100 cigarettes in his/her life?

**Table 2 biomedicines-11-00273-t002:** Values of cognitive impairment obtained for each cluster in terms of the MMSE.

MMSE						*p*-Value		
PPi	Cluster1	Cluster2	Cluster3	*WCSS*	*SSB*	Cluster 1 vs. 2	Cluster 1 vs. 3	Cluster 2 vs. 3
PP1	29.04 ± 0.89	28.27 ± 1.72	29.15 ± 0.86	26.364	18.430	0.021	0.118	0.000
PP2	29.05 ± 0.88	28.39 ± 1.45	29.13 ± 0.87	27.187	19.568	0.021	0.121	0.013
PP3	29.05 ± 0.81	28.39 ± 1.45	29.12 ± 0.84	27.337	18.112	0.074	0.143	0.011
PP4	28.91 ± 1.10	28.89 ± 1.16	28.96 ± 1.11	26.223	18.278	0.047	0.113	0.002
PP5	28.97 ± 0.81	28.28 ± 1.40	29.31 ± 0.59	26.029	19.341	0.012	0.093	0.000
PP6	28.87 ± 1.11	28.42 ± 1.40	29.29 ± 0.64	27.822	18.511	0.016	0.091	0.000
PP7	28.91 ± 1.75	28.90 ± 1.77	28.98 ± 1.38	31.796	15.180	0.129	0.106	0.043
PP8	28.91 ± 1.67	28.92 ± 1.61	28.98 ± 1.64	32.027	15.013	0.091	0.176	0.066
PP9	29.12 ± 1.24	28.39 ± 2.45	29.05 ± 1.31	33.011	14.372	0.089	0.192	0.073

**Table 3 biomedicines-11-00273-t003:** Comparison of demographics and relevant clinical variables among the three Aβ-PET phenotypes that emerged from PP5.

Clinical Variable	Parameter	Phenotype1	Phenotype2	Phenotype3
Gender	Female	96	70	136
	Male	98	48	70
Age	Mean ± std	67.30 ± 8.40	73.52 ± 5.83	62.12 ± 8.90
MMSE	Mean ± std	28.97 ± 0.81	28.28 ± 1.40	29.31 ± 0.59
Diabetes	Yes	16	93	12
	No	149	5	169
CDR	0	158	77	188
	0.5	26	24	17
	1	2	3	1
	2	0	2	0

Gender is provided as the number of males and females. Age and MMSE are shown as the mean and standard deviation. Diabetes is shown as a binary condition. CDR: Clinical Dementia Rating is shown as 0: no dementia; 0.5: questionable dementia; 1: mild cognitive impairment; 2: moderate cognitive impairment; and 3: severe cognitive impairment.

**Table 4 biomedicines-11-00273-t004:** Compilation of the total number of subjects in relation to some relevant epigenetic variables among the three Aβ-PET phenotypes determined for PP5.

Clinical Variable	Parameter	Phenotype1	Phenotipe2	Phenotype3
Education_years	mean ± std	16.010 ± 2.400	15.898 ± 2.553	16.247 ± 1.589
TOBAC	Yes	73	58	73
	No	88	38	107
DEPOTHR	Yes	24	19	5
	No	103	67	178
APOE	Low risk	134	60	122
	Medium risk	48	47	71
	High risk	5	11	5
Alcohol	Yes	3	7	4
	No	58	62	80

## Data Availability

The datasets analyzed during the current study are available in the XNAT repository, https://central.xnat.org/app/template/XDATScreen_report_xnat_projectData.vm/search_element/xnat:projectData/search_field/xnat:projectData.ID/search_value/OASIS3 (accessed on 10 July 2022).

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
