# Peer review of "Data-Driven Phenotyping of Alzheimer’s Disease under Epigenetic Conditions Using Partial Volume Correction of PET Studies and Manifold Learning"

_biomedicines, 2023, doi:10.3390/biomedicines11020273_

Round 1

Reviewer 1 Report

This paper proposed a new methodology to identify Alzheimer Disease (AD) phenotypes by selecting the best Phenotyping Pipeline under several epidemiological conditions. The method identifies 3 phenotypes and permits to analyze under epigenetic conditions. Experimental results on the OASIS-3 dataset demonstrate the effectiveness of the proposed model. Overall, this paper is well organized. Here are some specific comments for minor revision.

Q1In addition to p-value, the authors could consider including additional evaluation metrics in their analysis to further demonstrate the effectiveness of their method. Some examples of such metrics could include measures of accuracy, precision, recall, and F1 score for classification tasks, or mean squared error and coefficient of determination for regression tasks. By presenting a diverse range of evaluation metrics, the authors can provide a more comprehensive assessment of their method's performance and demonstrate its superiority over alternative approaches.

Q2For the evaluation index MMSE of different PPi used by the authors in 4.2, the authors should indicate the source of the method or give a detailed explanation of the method and the calculation process and the meaning of the expression to facilitate the reader's understanding.

Q3To strengthen the validity of the results obtained, we recommend that the authors consider conducting experiments on additional datasets in addition to the OASIS-3 dataset. This could include both public and non-public datasets to provide a more comprehensive evaluation of the proposed approach.

Q4: Previous works have discussed how brain MR to PET synthesis and brain MR image super-resolution can improve AD diagnosis (Bidirectional mapping generative adversarial networks for brain MR to PET synthesis;  Fine perceptive gans for brain mr image super-resolution in wavelet domain, DOI:10.1109/TNNLS.2022.3153088). It is better to discuss the influence. Besides, AD recognition using brain image and machine learning is a hot topic. It is better to give a more extensive review. There are many new works in this field. such as:

Tensorizing GAN with high-order pooling for  Alzheimer's disease assessment;

Automatic recognition of mild cognitive impairment from MRI images using expedited convolutional neural networks.

Q5 The Figure 1 is blurrier when enlarged, so authors are invited to insert vector images in the text, such as eps, pdf format. Table 2 is very wide in layout, and the authors should unify the table format to make the article look neater and more beautiful as a whole.

Q6The paper is well organized generally. However, there are several grammatical errors in different parts of the text. For example, on the first page, the penultimate five lines of no treatments of AD … should be changed to “… no treatments for AD”. The “take advantage to” in the second line of the second page should be changed to “take advantage of”, which would be more appropriate.

Author Response

Reviewer1

Thank you very much for your comments, questions and suggestions, all have been taken into consideration. See the following lines and answers with reference to the corrected manuscript to track the changes. In black your original questions, in red our answers, and the new manuscript has change track to follow the progress.

This paper proposed a new methodology to identify Alzheimer Disease (AD) phenotypes by selecting the best Phenotyping Pipeline under several epidemiological conditions. The method identifies 3 phenotypes and permits to analyze under epigenetic conditions. Experimental results on the OASIS-3 dataset demonstrate the effectiveness of the proposed model. Overall, this paper is well organized. Here are some specific comments for minor revision.

Q1In addition to p-value, the authors could consider including additional evaluation metrics in their analysis to further demonstrate the effectiveness of their method. Some examples of such metrics could include measures of accuracy, precision, recall, and F1 score for classification tasks, or mean squared error and coefficient of determination for regression tasks. By presenting a diverse range of evaluation metrics, the authors can provide a more comprehensive assessment of their method's performance and demonstrate its superiority over alternative approaches.

We acknowledge the reviewer for the comment. We have already considered how to present additional evaluation metrics that would provide a more comprehensive assessment of the method's performance and demonstrate its superiority over alternative approaches. The issue here is that the usual metrics involved in measuring artificial intelligence classifiers (namely measures of accuracy, precision, recall, and F1 score) in our case are not eligible as they require having a value of real trues and false in order to build the confusion matrix elements (TP, TN, FP and FN). In our case there is not the true phenotypes, only possible values provided by different classifiers are obtained. In any case, as the recommendation is good, we have included a sentence on the discussion indicating the convenience of producing a new future article exploring these possibilities. See lines 611-620.

Q2For the evaluation index MMSE of different PPi used by the authors in 4.2, the authors should indicate the source of the method or give a detailed explanation of the method and the calculation process and the meaning of the expression to facilitate the reader's understanding.

We thank you very much for this question. You are right, and we agree the comment. A new paragraph including the required explanation of generation of MMSE indexes, see lines 475-482 to check the added explanation. The following paragraph have been corrected.

Once the three clusters are identified for all PPi, the values of cognitive impairment have been calculated by computing the mean value and the standard deviation of patients' Mini–Mental State Examination (MMSE) included on each cluster, allowing the phenotype identification and characterization. MMSE measures general cognitive status, ranging from 0 (severe impairment) to 30 (no impairment), based on a 30-point questionnaire that is used extensively in clinical and research settings [19]. Results for each PPi are provided in Table 2. Furthermore, this table also provides the p-value of the inter phenotypes comparisons, using one-way analysis of variance (ANOVA) test, as well as the intra cluster and inter cluster measurements, using Equations (4 and 5).”

Q3To strengthen the validity of the results obtained, we recommend that the authors consider conducting experiments on additional datasets in addition to the OASIS-3 dataset. This could include both public and non-public datasets to provide a more comprehensive evaluation of the proposed approach.

Thank you very much for the comment, this is a valuable point. Authors take the point into consideration. A paragraph in lines 603-609 has been add on the new manuscript.

The inclusion of another databases (public as ADNI) or private (from our hospital or any other research group that could provide it) could be a way to provide more robust results. On the other hand, it is important to take into consideration two important facts, a) the need of having simultaneously both kinds of imaging studies (PET and MRI) and the clinical variables employed on the studio. b) this is an additional analysis that lies out of the scope of this paper and is a clear new future work line.

So, we take into consideration of your suggestion as a new future work line, as we consider that this additional analysis lies out of the scope of this paper.

Q4: Previous works have discussed how brain MR to PET synthesis and brain MR image super-resolution can improve AD diagnosis (Bidirectional mapping generative adversarial networks for brain MR to PET synthesis; Fine perceptive gans for brain mr image super-resolution in wavelet domain, DOI:10.1109/TNNLS.2022.3153088). It is better to discuss the influence. Besides, AD recognition using brain image and machine learning is a hot topic. It is better to give a more extensive review. There are many new works in this field. such as:

Tensorizing GAN with high-order pooling for Alzheimer's disease assessment;

Automatic recognition of mild cognitive impairment from MRI images using expedited convolutional neural networks.

Thanks for the comment, we have also included a new paragraph discussing the influence, where you can see the paragraph:

In addition to PET studies, Magnetic Resonance Imaging (MRI) have also been extensively employed to assess AD. Even if MRI usage is limited by factors such as imaging hardware, scanning time, and costs, this imaging modality plays an important role in clinical and brain exploration [14]. Previous works have discussed their role on cognitive impairment assessment from different approaches, as using functional MRI, acquiring high-resolution MRI images clinically using super-resolution [15] or how brain MRI to PET synthesis can improve AD diagnosis. In this paper T1 MRI studies are employed to build the brain atlas for each patient.”

See lines 85-92 on the new manuscript. And the additional references [14, 15].

Q5 The Figure 1 is blurrier when enlarged, so authors are invited to insert vector images in the text, such as eps, pdf format. Table 2 is very wide in layout, and the authors should unify the table format to make the article look neater and more beautiful as a whole.

Thanks for the comment, we have taken into consideration the suggestion and we have also included a new Figure 1 and Table “following the recommendations, See lines 182 and 483 on the new manuscript

Q6The paper is well organized generally. However, there are several grammatical errors in different parts of the text. For example, on the first page, the penultimate five lines of “… no treatments of AD …” should be changed to “… no treatments for AD”. The “take advantage to” in the second line of the second page should be changed to “take advantage of”, which would be more appropriate.

Thanks for the comments, we have already corrected in the manuscript. See lines 42 and 48 for the corrected versions.

Reviewer 2 Report

Thank you for your submissions. It is a good manuscript. Congratulations to your work.

I don't quite understand why unsupervised (hence the use of k-means) is emphasized in clustering. Supervised and fuzzy clustering has been well studied and implemented on PET.

Minors:

The expression is fluent, but there are still some typos, such as.

123 an extra comma.

270 the PET Unified Pipeline

301 PPi-s. And there should be a comma before.

364 numerically -> statistically

390 determination k?

404 the largest

432 A new paragraph should not start with "this"

447 a to i

Most superscripts are unnecessary. As long as you explain the abbreviations in the caption, they don't have to be superscripted.

LDA can be mentioned in the introduction of dimensionality reduction, which is also widely used in PET studies

Author Response

Reviewer2

Thank you very much for your comments, all have been taken on consideration. See the following lines and answers with reference to the corrected manuscript to track the corrections. In black your original questions, in red our answers, and the new manuscript has change track to follow the progress.

Thank you for your submissions. It is a good manuscript. Congratulations to your work.

I don't quite understand why unsupervised (hence the use of k-means) is emphasized in clustering. Supervised and fuzzy clustering has been well studied and implemented on PET.

In the approach presented in this work, the employment of unsupervised methods is mandatory. The above-mentioned supervised approaches require to know, a priori, to which phenotype belong each subject (label information to train the algorithms) that is not known in our case during the internal stages of the process. And that is why the incorporation of a non-supervised approach (in our work k-means) makes things possible.

Minors:

The expression is fluent, but there are still some typos, such as.

123 an extra comma.

Now is Corrected, now it can be seen in line 121,

270 the PET Unified Pipeline

Now is Corrected, now it can be seen in line 279,

301 PPi-s. And there should be a comma before.

Now is Corrected, now it can be seen in line 311,

364 numerically -> statistically

Now is Corrected, now it can be seen in line 374,

390 determination k?

Now is Corrected, now it can be seen in line 400,

404 the largest

Now is Corrected, now it can be seen in line 411,

432 A new paragraph should not start with "this"

Now is Corrected, now it can be seen in line 439

447 a to i

Now is Corrected, now it can be seen in line 455

Most superscripts are unnecessary. As long as you explain the abbreviations in the caption, they don't have to be superscripted.

Yes, thanks for the comment, all superscripts have been removed. Now, once corrected, the new rewriting can be seen in lines 383-389, 493-497 and 516-520.

LDA can be mentioned in the introduction of dimensionality reduction, which is also widely used in PET studies

Now is Corrected, now it can be seen in lines 97-98 and 192.
